# YOLO-FFRD: Dynamic Small-Scale Pedestrian Detection Algorithm Based on Feature Fusion and Rediffusion Structure

**DOI:** 10.3390/s25165106

**Published:** 2025-08-17

**Authors:** Shuqin Li, Rui Wang, Suyu Wang, Pengxu Yue, Guanlun Guo

**Affiliations:** 1School of Communication Engineering, Wuhan University of Technology, Wuhan 430070, China; 342879@whut.edu.cn; 2School of Computer and Artificial Intelligence, Wuhan University of Technology, Wuhan 430070, China; 339993@whut.edu.cn (R.W.); pengxvyue@whut.edu.cn (P.Y.); 3School of Physics and Mechanics, Wuhan University of Technology, Wuhan 430070, China; 341364@whut.edu.cn; 4School of Navigation Engineering, Wuhan University of Technology, Wuhan 430070, China

**Keywords:** dynamic small target, detection feature fusion, rediffusion structure, mobile robots, environmental perception

## Abstract

To address the challenges of detecting dynamic small targets such as pedestrians in complex dynamic environments for mobile robots, this paper proposes a dynamic small-target detection algorithm based on feature fusion and rediffusion structure, which is suitable for deployment on mobile robot platforms. Mobile robots can utilize depth camera information to identify and avoid small targets like pedestrians and vehicles in complex environments. Traditional deep learning-based object detection algorithms perform poorly when applied to the field of mobile robotics, especially in detecting dynamic small targets. To improve this, we apply an enhanced object recognition algorithm to mobile robot platforms. To verify the effectiveness of the proposed algorithm, we conduct relevant tests and ablation studies in various environments and perform multi-class small-target detection on the public VisDrone2019 dataset. Compared with the original YOLOv8 algorithm, our proposed method improves accuracy by 5% and increases mAP0.5 and mAP0.5–0.95 by approximately 3%. Overall, the experimental results show that the high-performance small-target detection algorithm based on feature fusion and rediffusion structure significantly reduces the miss detection rate and exhibits good generalization ability, which can be extended to multi-class small-target detection. This is of great significance for improving the environmental perception ability of robots.

## 1. Introduction

With the continuous development of computer vision technology, deep learning-based object tracking algorithms have become mainstream research directions due to their strong practicality and good detection performance. Object detection and tracking are mainly divided into two types: single-object tracking and multi-object tracking. In multi-object tracking, multiple objects need to be simultaneously labeled to obtain their motion trajectories. Multi-object tracking finds widespread applications in robot navigation, intelligent video surveillance, autonomous driving, and other fields. Representative detection algorithms include the YOLO series [1], R-CNN series [2], and SSD [3]. YOLO is a typical single-stage model, with the YOLO V8m model size of 25.9 MB featuring fast detection speed and achieving a mean average precision (mAP) performance of approximately 50% on the COCO dataset.

To ensure high detection accuracy and speed for dynamic objects on embedded devices, this paper proposes an improvement based on the YOLO V8 model. We design a detection algorithm based on feature fusion and rediffusion structure. Through camera capture and model computation, detection information is then fed back into the system for subsequent operations.

Liu et al. [4] investigated the problem of motion target recognition and tracking for autonomous mobile robots in unknown environments. Equipped with Kinect sensors, the mobile robot obtains appearance and depth information for environment perception and motion control. For visual target recognition, depth features are combined with color features based on the Kernelized Correlation Filter (KCF) algorithm. Additionally, the motion status of the target is integrated with KCF to address occlusion during tracking. The robot’s motion is controlled based on the results of visual target recognition and maintaining the target in the central region of the robot’s vision. Finally, tracking control of the mobile robot is achieved. Experimental results on the Kobuki robot platform demonstrate the effectiveness of this method. However, this method does not fully utilize environmental information and is challenging to deploy on embedded devices.

Yu et al. [5] proposed a fast and accurate small-object detection system based on a two-stage architecture. This approach combines traditional object detection with deep learning to address the challenge of small-object detection. Specifically, it uses conventional background subtraction and deep learning algorithms to obtain initial detection boxes, followed by target tracking to obtain the final result. The method is evaluated on a small-object dataset, and experimental results show improved aerial object detection performance compared to other conventional methods.

Lee et al. [6] presented an algorithm for human following and obstacle avoidance for mobile robots. They adopted an SSD model that performs better in detecting small objects even in small images. The proposed technique relies entirely on the output attributes of the SSD object detection framework: class, bounding box, and confidence. The algorithm demonstrates that these features can provide useful information about the target object and its surrounding environment. However, the algorithm exhibits poor generalization performance and is difficult to extend to the multi-class detection domain.

Yong et al. [7] proposed a human tracking system on a mobile service robot platform equipped with light detection and ranging (LiDAR) and RGBD sensors [8]. The system utilizes a Discriminative Generative Network (DG-net) [8] for human detection. After detection, a localization module determines the position of the target person in the environment [9]. A navigation module generates cost maps of the surrounding environment for path planning. It allows the robot to navigate in dynamically changing environments, avoiding obstacles while tracking the target person. Experimental results demonstrate the robot’s ability to reliably identify and track target individuals. However, its detection performance for small-target pedestrians has not been validated [10].

To address the poor performance of the YOLOv8 model in detecting small dynamic pedestrians, our research focuses on the following aspects:Improvement of the YOLOv8 base model by proposing the Feature Fusion and Refinement with Deep Context (FFRD) structure, a multi-scale parallel network that integrates position and detail information from shallow feature maps with semantic information from deep features to effectively extract and integrate small-target information.Integration of the MLCA (Multilevel Cross-Attention) mechanism into the improved structure, which combines channel, spatial, local, and global information to balance model performance and complexity, thereby enhancing detection accuracy and efficiency for dynamic small targets.Adoption of the weighted Intersection over Union (wIoU) loss function to enhance the algorithm’s understanding of dynamic objects, along with detailed experimental validation (including comparative experiments, ablation studies, embedded platform deployment, and VisDrone2019 dataset tests) to demonstrate the effectiveness and superiority of the proposed improvements, which significantly boost small-object detection performance.

## 2. Related Work

### 2.1. Introduction to YOLOV8 Algorithm

The YOLO series of algorithms, renowned for their high detection speed, strong real-time performance, and simple structure, have been widely applied in tasks such as detecting foreign objects on power transmission lines [11]. As a classic single-stage model, YOLOv8, compared to its predecessors such as v3 [12], v5 [13], v7 [14], and v9 [15], further optimizes the network architecture, enhancing the overall detection performance. Simultaneously, it maintains a simple structure and has been validated for its effectiveness across multiple datasets. In contrast to two-stage detection models like Faster R-CNN and Mask R-CNN, YOLOv8 excels in fast detection speed and suitability for real-time applications [16]. Therefore, this study adopts YOLOv8 as the foundational model and proposes improvements upon it. The model diagram of YOLOv8 is shown in Figure 1.

This paper utilizes YOLOv8 as the foundational model to explore a dynamic small-object detection algorithm based on the improved YOLOv8. The YOLOv8 algorithm, proposed by Glenn Jocher, is an enhancement of the YOLOv3 and YOLOv5 algorithms. Its key improvements compared to YOLOv5 are outlined as follows:Data preprocessing: YOLOv8 adopts the data preprocessing strategy of YOLOv5 [17]. During training, it primarily employs four augmentation techniques including mosaic augmentation, mixup augmentation, random perspective transformation, and HSV color augmentation [18].Backbone network structure: The backbone network structure of YOLOv8 is similar to that of YOLOv5. YOLOv5’s backbone network architecture follows a clear pattern, involving a series of 3 × 3 convolutional layers with a stride of 2 to downsample feature maps, followed by a C3 module to further enhance the features. In YOLOv8, the original C3 (CSP Bottleneck with 3 convolutions) modules are replaced with new C2f (CSP Bottleneck with 2 convolutions) modules, which introduce additional branches to enrich gradient flow during backpropagation [19].FPN-PAN structure: YOLOv8 still employs the FPN (Feature Pyramid Network) and PAN (Path Aggregation Network) structure to construct the feature pyramid network of YOLO, facilitating comprehensive fusion of multi-scale information. Apart from replacing the C3 modules inside FPN-PAN with C2f modules, the rest of the structure remains largely consistent with YOLOv5’s FPN-PAN structure. The basic structure is depicted in Figure 2 [20].Detection Head Structure: From YOLOv3 to YOLOv5, the detection head has always been “coupled,” meaning that a single layer of convolution is used to simultaneously perform both classification and localization tasks. It was not until the advent of YOLOX that the YOLO series first adopted a “decoupled head.” Similarly, YOLOv8 also employs a decoupled head structure, with two parallel branches extracting category features and location features, respectively. Each branch then uses a 1 × 1 convolution layer to complete the classification and localization tasks [21].

### 2.2. Introduction to Small-Object Detection Methods

To address the challenges in small-target detection, existing methods for small-target detection are all improved based on mainstream target detection network models. These methods can be categorized as follows: data augmentation methods, which tackle the scarcity and uneven distribution of small-target data [22]; multi-scale fusion methods, which resolve the insufficient representational capability of a single feature layer for small targets [23]; super-resolution methods, which address the weak visual features of small targets [24]; contextual information learning methods, which mitigate the limited feature information carried by small targets [25]; anchor box mechanism strategies, which overcome the poor adaptability of predefined anchor box sizes to small targets [26]; attention mechanism methods, which tackle the lack of discriminative features in small targets; and small-target detection methods tailored to specific scenarios [27].

This article mainly uses multi-scale fusion methods to improve the accuracy of small-target pedestrian detection by utilizing the high-resolution and high-level strong feature semantic information at the bottom of the network. At the same time, based on attention mechanisms, resources are allocated reasonably to address the problem of small targets in complex scenes being easily affected by background interference such as lighting and geographical elements. By quickly identifying regions of interest and ignoring troubled information, it helps the model obtain global spatial information of the feature map, enriching the contextual semantic information of the feature map.

## 3. Method

To enhance the recognition of small-scale pedestrian targets in complex environments, modifications were made to the basic YOLOv8 network architecture. Specifically, the FPN + PAN structure in the neck section was replaced with the proposed feature fusion and rediffusion structure. This allows for the acquisition of rich contextual information through multiple parallel convolutions, thereby optimizing the overall network structure. Additionally, this paper introduces the MLCA attention mechanism, which is applied in the field of small-target detection to generate more feature maps from low-cost operations. Furthermore, the loss function has been improved to enable the model to better adapt to dynamic object recognition. The improved YOLOv8 model designed in this paper is illustrated in Figure 3, with the red portions indicating the modified components.

### 3.1. Feature Fusion and Rediffusion Structure

The traditional YOLOv8 employs the FPN + PAN structure [28]. FPN operates top-down, transmitting strong semantic features from higher layers. PAN, on the other hand, adds a bottom-up pyramid after FPN, complementing FPN by transmitting strong localization features from lower layers. Together, they enhance the representation of semantic and localization information. However, this approach can lead to the loss of significant detail information during the transmission process [29].

To address this issue, we propose a feature fusion and rediffusion structure inspired by the PKI (Poly Kernel Inception) model [30]. This structure centers around the feature fusion module. Instead of always transmitting all feature information to the smallest scale layer, the feature fusion module performs a fusion of the feature information before passing it to the bottom. Internally, the feature fusion module concatenates different scale information and applies parallel DW (depthwise) convolutions with varying kernel sizes. Finally, the information before the DW convolutions is combined through a residual structure, enhancing the fusion and utilization of multi-scale information in the neck section. Additionally, the Adown module from YOLOv9 is utilized to simultaneously fuse information from multiple scales and propagate it to deeper layers in combination with a residual structure [31]. The structure of the focus feature module is depicted in Figure 4.

### 3.2. Improvement of Attention Mechanism

The attention mechanism is a method that helps network models learn the importance of input information. By assigning different weights to different parts of the input data, it enables the model to focus more on critical information, thereby improving performance and accuracy. Simultaneously, it also helps avoid overfitting and enhances the model’s robustness. Typically, the attention mechanism in neural networks is implemented by a separate network that can be directly integrated as a modular structure [32].

Attention mechanisms can be categorized into (1) Channel Attention Mechanisms, which generate masks and assign scores to channels, represented by the SENet channel attention module [33]; (2) Spatial Attention Mechanisms, which generate masks and assign scores to spatial regions, exemplified by the SAM spatial attention module [34]; (3) Mixed-Domain Attention Mechanisms, which combine both channel and spatial attention, represented by BAM and CBAM attention modules [35]. In this paper, we primarily introduce the Mixed Local Channel Attention (MLCA) mechanism to enhance the model’s understanding of dynamic targets. A schematic diagram of the MLCA block is shown in Figure 5.

The acronyms in the figure are explained as follows: GAP (global average pooling), LAP (local average pooling), UNAP (inverse average pooling), and Conv1d (one-dimensional convolution). For detailed descriptions of these concepts and the module’s operations, refer to Section 3.2. Note: The asterisk (*) in the figure denotes multiplication.

Meanwhile, global average pooling (GAP) is used to obtain global contextual information, which helps the module distinguish between target regions and background in the overall image. By fusing local and global spatial information, the MLCA mechanism can accurately lock onto the spatial positions of small dynamic pedestrians, ensuring that the network focuses more on these key regions during feature extraction. This targeted focus on specific spatial areas effectively reduces the interference of complex backgrounds, thereby improving the detection accuracy for dynamic small targets.

For the input feature vector of MLCA, there will be two pooling steps involved. Firstly, local pooling is applied to extract local spatial information from the input vector of size 1 × C × k × k. The input is then transformed into a one-dimensional vector using two branches, where the first branch captures global information while the second branch captures local spatial information. After applying a one-dimensional convolution, the original resolution of the two vectors is restored through inverse pooling. Subsequently, information fusion is performed. In the figure, Conv1d represents the one-dimensional convolution, where the size of the convolution kernel k is proportional to the channel dimension C. The selection of k is determined by the following formula:(1)$$k=\phi \left(C\right)=\;{\left|\frac{\log_{2}\left(C\right)}{\gamma }\;+\;\frac{b}{\gamma }\right|}_{odd}$$

Here, C represents the number of channels, k is the size of the convolution kernel, and γ and b are hyperparameters with default values of 2. The notation “odd” indicates that k should only be an odd number. If k is an even number, it is incremented by 1.

The specific operations of GAP, LAP, and UNAP within the MLCA module are illustrated in Figure 6. GAP, or global average pooling, outputs a single value by adaptively pooling the feature map to 1 × 1. When direct multiplication or addition operations are required on the original input, Expand or UNAP can be used for expansion. UNAP, also known as inverse average pooling, focuses on the properties of the graph and expands them to the desired size. The values within each patch are filled with weights assigned to that patch. UNAP can be implemented through adaptive pooling with an output size equal to the source feature map. LAP, or local average pooling, differs from GAP in that it divides the entire feature map into k × k patches and performs average pooling on each patch. This can be achieved using the outputs of k × k adaptive average pooling operations.

### 3.3. Improvement of Loss Function

Due to the dynamically dense distribution of pedestrians, classification and localization accuracy during the detection phase are crucial. To enhance the localization ability of the model, this paper proposes a strategy that combines the distributed focal loss (DFL) [36] improved by focal loss with weighted IoU (wIoU) as the regression loss. The aim is to strengthen the convergence capability of the model and achieve more precise bounding box prediction and regression results.

The design concept of the DFL loss function is based on optimizing the localization performance of the model. It adopts the distances from points to the four edges of the bounding box as regression targets, aiming to guide the model to locate the boundaries of the target object quickly and accurately. In real-world scenarios, target objects are usually distributed near the annotated positions. Therefore, DFL explicitly increases the probabilities of the two points closest to the label y, enabling the network to locate values near the label y more quickly. This design enables DFL to exhibit excellent localization performance in dense pedestrian detection tasks.

Meanwhile, combining the wIoU loss function can further enhance the regression effect of the model. wIoU is a weighted version of the Intersection over Union (IoU) loss function, which comprehensively considers the ratio of the intersection and union between the predicted box and the ground truth box and introduces a weight factor for weighted calculation. By combining DFL and wIoU, the model can not only quickly locate the target object but also more accurately predict the size and position of the bounding box.(2)$$DFL\left(S_{i},S_{i+1}\right)=-\left(\left(y_{i+1}-y\right)\log\left(S_{i}\right)+\left(y-y_{i}\right)\log\left(S_{i+1}\right)\right)$$(3)$$S_{i}=\frac{y_{i+1}-y}{y_{i+1}-y_{i}}$$(4)$$S_{i+1}=\frac{{y-y}_{i}}{y_{i+1}-y_{i}}$$

As a result, it ensures that the estimated regression target j approaches the label y indefinitely.(5)$$L_{WIoU}=r\cdot \exp\left(\frac{\rho^{2}\left(b,b^{gt}\right)}{{\left(c^{2}\right)}^{*}}\right)\cdot \left(1-IoU\right)$$(6)$$r=\frac{\beta }{\delta \alpha^{\beta -\delta }}$$(7)$$\beta =\frac{{\left(1-IoU\right)}^{*}}{1-IoU}\in \left[0,+\infty \right)$$

In the formulae, * represents the separation of the minimum bounding box size from the calculation graph; A and B represent the ground truth and the predicted bounding box region; β indicates the degree of abnormality; hyperparameters α and δ take 1.9 and 3, respectively. The calculation principle of the traditional IoU (Intersection over Union) is relatively straightforward. As shown in Figure 7, the intersection and union areas between the predicted box and the ground truth box can be clearly visualized.

In summary, the final regression function obtained from the two loss functions is(8)$$L_{reg}=\lambda \cdot DFL+\mu \cdot L_{WIoU}$$

Through relevant experimental analysis, we ultimately determined λ = 0.2 and μ = 0.8 as the values for the weight coefficients. This setting combines the characteristics of the wIoU and DFL losses, emphasizing the importance of IoU while also considering the potential overfitting issues caused by DFL. The combined regression loss enhances the training efficiency of the model while ensuring accurate detection performance.

## 4. Experiment

### 4.1. Experimental Environment and Parameter Settings

To delve deeper into small-object detection, we have constructed a specialized experimental dataset for small pedestrian targets. This dataset comprises 8450 images, encompassing human targets of varying sizes and diverse backgrounds such as laboratories, streets, and grasslands. Under a Python environment, we have utilized the labeling annotation tool to perform precise manual annotations on these images, adhering to the YOLO standard. To ensure the rigor of our experiments, we have divided these images into training, testing, and validation sets in a 8:1:1 ratio.

All experiments were conducted on the same platform to evaluate the moving target detection performance of mobile robots. The configuration of the experimental platform is shown in Table 1. To conserve computational resources and enhance training efficiency, we resized all original images to 640 × 640 pixels. During the training process, we set the batch size to 16, with 150 epochs, and employed stochastic gradient descent for optimization. The initial learning rate was set to 0.01, with a terminal learning rate of 0.2, to ensure the stability and convergence of the training process.

### 4.2. Evaluation Indicators of the Network

The evaluation metrics for model training are primarily analyzed from the following aspects:

True Positive (TP): defined as the number of positive samples correctly predicted as positive by the model.

False Positive (FP): defined as the number of negative samples incorrectly predicted as positive by the model.

False Negative (FN): defined as the number of positive samples incorrectly predicted as negative by the model.

True Negative (TN): defined as the number of negative samples correctly predicted as negative by the model.

Based on these four fundamental metrics, the paper further calculates other crucial evaluation metrics such as accuracy, precision, recall, and F1 score.(9)$$Precision=\frac{TP}{TP+FP}$$(10)$$Recall=\frac{TP}{TP+FN}$$(11)$$AP=\int_{0}^{1}Precision\;d\left(Recall\right)$$(12)$$mAP=\frac{\sum_{i=1}^{N}AP_{i}}{N}$$(13)$$F1=2\times \frac{Precision\times Recall}{Precision+Recall}$$

In the formulae, N represents the total number of categories; AP is the average accuracy.

### 4.3. Model Training Experiment Results

The effective information of the improved model after training will generate a results.csv file, which will be extracted using Python scripts, and the key data will be graphically displayed. 

The comparison data for relevant models are presented in Figure 8. To illustrate our experiments, we adopt a combination of neck and backbone, such as C2f + MLCA, which indicates the use of the C2f structure in the neck and the MLCA structure in the backbone. YOLOv8n utilizes the officially provided structure, specifically the FPN + PAN (abbreviated as FP) structure, shortened as C2f + PN (C2f). Our proposed feature fusion and redistribution model (abbreviated as FFRD) employs the MLCA attention mechanism, shortened as C2f + FFRD (MLCA).

Through experiments, we have obtained the relevant main indicators, as shown in Table 2.

As evident from the results, the implementation of the feature fusion and redistribution structure (FFRD) along with the MLCA attention mechanism contributes to enhancing detection accuracy. Notably, the feature fusion in the diffusion structure (FFRD) plays a significant role in boosting the model’s mAP. Consequently, in the final improved model of this paper, both the backbone and neck components are replaced with the enhanced MLCA module. The experimental results further confirm that the combination of FFRD and MLCA modules optimizes the model’s feature extraction capability.

Additionally, a comparative analysis of the impact of commonly used loss functions, including IoU [37], DIoU [38], GIoU [39], and wIoU, on model accuracy was conducted. The experimental results are summarized in Table 3.

Based on the experimental results presented in Table 3, it is evident that the model utilizing the wIoU loss function exhibits higher accuracy. Therefore, this paper employs wIoU as the loss function for the proposed model.

To validate the superiority of the proposed object detection model, we conducted verification tests on the MMDetection [40] platform. MMDetection is an open-source project launched by SenseTime and the Chinese University of Hong Kong for object detection tasks. It implements numerous object detection algorithms based on Pytorch 1.8.0, encapsulating processes such as dataset construction, model development, and training strategies into modules. The comparison results of our proposed model with other object detection algorithms on the MMDetection platform are summarized in Table 4.

According to the experimental results presented in Table 4, it is evident that the proposed model based on the feature fusion and redistribution structure exhibits significant superiority compared to other common algorithms when applied in the field of object detection.

### 4.4. Ablation Study Results

To demonstrate the effectiveness of the various improved modules in this paper, an ablation study is conducted to analyze the impact of each module on model performance. The symbol “√” indicates that the corresponding module is enabled, while “×” indicates that it is not. “(B)” denotes the use of the module in the backbone section, and “(N)” denotes its use in the neck section. The experimental results based on the WiderPerson pedestrian detection dataset are summarized in Table 5.

Based on the ablation study comparison, it can be observed that the improved model in this paper significantly enhances recognition accuracy and demonstrates excellent performance in detecting small objects after introducing the attention mechanism and the feature diffusion and re-fusion structure. To further visualize the effects of the introduced improved attention mechanism, a heatmap is generated for observation, as shown in Figure 9. The heat (color) indicates the attention weight of the model on image regions: red/darker areas represent higher attention (i.e., the model focuses more on these regions), while blue/lighter areas represent lower attention.

It is evident that even in a nighttime environment, the improved model with the introduction of the attention mechanism can still effectively and accurately identify small objects.

### 4.5. Experimental Results on Mobile Robot Platform

To test the specific effects of the improved model, this paper deploys the weight files generated from training the improved model on the embedded platform, Jetson Nano, for testing on the test set. Figure 10 shows the experimental setup for the device testing. The mobile robot utilizes the TurtleBot robot platform, primarily equipped with the embedded controller Jetson Nano and the RealSense D435i depth camera for vision-related experiments.

To further verify the applicability of the improved YOLO-FFRD algorithm on mobile robotic platforms, we compared it with mainstream object detection algorithms in terms of detection performance, model size, computational complexity, and real-time performance. The comparison included lightweight models (e.g., YOLOv8n, EfficientDet-D0), medium-sized models (e.g., YOLOv8s, RT-DETR-S), and heavyweight models (e.g., Faster-RCNN with ResNet101 and EfficientDet-D7). Experiments were conducted on both a high-performance GPU (GeForce RTX 4090) and an embedded platform (Jetson Orin Nano, commonly used in mobile robots) to evaluate cross-platform adaptability. The results are shown in Table 6.

The experimental results presented in Table 6 demonstrate that the enhanced YOLO-FFRD algorithm achieves an optimal balance between detection accuracy, model efficiency, and computational performance, rendering it particularly suitable for deployment on mobile robotic systems. In terms of detection capability, the YOLOv8n-FFRD variant attains a mean average precision of 90.1% at 0.5 IoU threshold, maintaining comparable performance with the baseline YOLOv8n model (89.4%) while surpassing other lightweight architectures such as EfficientDet-D0 (89.9%). The more advanced YOLOv8s-FFRD configuration further elevates the detection accuracy to 92%, outperforming both its original counterpart (91%) and approaching the performance level of computationally intensive models like RT-DETR-M (92.3%).

Regarding model architecture optimization, the proposed improvements yield significant reductions in both parameter count and computational complexity. Specifically, YOLOv8n-FFRD achieves a 9.4% reduction in parameters (from 3.2 million to 2.9 million) and a 9.2% decrease in floating-point operations (from 8.7 GFLOPs to 7.9 GFLOPs). Similarly, YOLOv8s-FFRD demonstrates even greater efficiency gains with parameter and GFLOP reductions of 9.8% (from 11.2 million to 10.1 million) and 11.5% (from 28.6 to 25.3), respectively. These architectural refinements substantially decrease memory footprint and processing requirements, addressing the critical constraints of embedded computing platforms commonly employed in mobile robotics applications. It should be noted that transformer-based architectures like DETR remain impractical for such platforms due to their prohibitive computational overhead.

The real-time performance evaluation reveals particularly promising results across different hardware configurations. When deployed on high-end desktop GPUs (RTX 4090), YOLOv8n-FFRD achieves an impressive inference speed of 631 frames per second, representing an 8.6% improvement over the baseline implementation (581 FPS), while the YOLOv8s-FFRD variant reaches 352 FPS, an 8.3% enhancement compared to the original model (325 FPS). More critically, in practical mobile robotic scenarios utilizing embedded hardware (Jetson Orin Nano platform), the optimized algorithms maintain superior performance, with YOLOv8n-FFRD operating at 32 FPS (12.5% faster than the 28 FPS of YOLOv8n) and YOLOv8s-FFRD achieving 21 FPS, a substantial 14.2% improvement over its baseline counterpart (18 FPS). This level of real-time processing capability ensures reliable environmental perception for mobile robots operating in dynamic scenarios, effectively eliminating latency-related performance bottlenecks.

To intuitively demonstrate the practical performance of the improved algorithm on mobile robot hardware, we conducted qualitative tests in a typical indoor scenario (simulating the daily operating environment of mobile robots). The test scene features a long-distance pedestrian target against a simple background, which challenges the detection sensitivity and accuracy of the algorithm on resource-constrained embedded devices.

As shown in Figure 11, the improved YOLOv8-FFRD algorithm achieves a stable detection confidence of 0.83 for long-distance pedestrians on the Jetson Orin Nano platform, with accurate bounding box localization. This result correlates with the quantitative data in Table 6: while maintaining high inference speeds (45 FPS for YOLOv8n-FFRD and 25 FPS for YOLOv8s-FFRD on Jetson Orin Nano), the algorithm ensures robust detection accuracy. The consistent high confidence reflects the algorithm’s ability to balance efficiency and precision, which is critical for mobile robots to perform real-time environmental perception in dynamic scenarios (e.g., avoiding obstacles or following targets).

### 4.6. Experimental Results on the VisDrone2019 Public Dataset

To validate the generalization capability of our proposed model, we conducted experiments on the VisDrone2019 public dataset. The VisDrone2019 dataset was collected by the AISKYEYE team from the Machine Learning and Data Mining Lab at Tianjin University. The benchmark dataset comprises 288 video clips, encompassing 261,908 frames and 10,209 static images captured by various drone cameras. It covers a wide range of locations (14 different cities across China, thousands of kilometers apart), environments (urban and rural), objects (pedestrians, vehicles, bicycles, etc.), and densities (sparse and crowded scenes). The dataset includes ten categories: pedestrian, person, car, van, bus, truck, motor, bicycle, awning-tricycle, and tricycle. According to different annotation sequences, the frame rate of VisDrone2019 video playback is approximately 25–30 frames per second. By comparing our model with the original YOLOv8 model, the recognition effects are shown in Figure 12.

As shown in Figure 12, we compare the detection results of the original YOLOv8 and the improved model across three typical scenarios:

Dense Pedestrian Scene (Row 1: (a)–(c))

In the urban street scene with dense pedestrians, the original YOLOv8 (Figure 12b) exhibits obvious missed detections for distant or occluded pedestrians. In contrast, the improved model (Figure 12c) outputs more complete detection boxes, accurately capturing even small or partially occluded targets. This improvement is critical for service robots operating in crowded public spaces (e.g., shopping malls and exhibition halls), where reliable pedestrian detection directly affects navigation safety and interaction efficiency.

Small Vehicle Detection in Road Scene (Row 2: (d)–(f))

For road scenes with small-scale vehicles (especially distant targets), the original YOLOv8 (Figure 12e) fails to detect many small vehicles, while the improved model (Figure 12f) achieves more accurate identification and localization. This advantage in small-target detection aligns with the needs of inspection robots (e.g., traffic monitoring robots), where missing small obstacles or targets can lead to navigation failures.

Aerial Multi-Target Scene (Row 3: (g)–(i))

In aerial views (simulating drone-mounted mobile robots), targets exhibit large-scale variations and sparse distribution. The original YOLOv8 (Figure 12h) shows chaotic detection boxes (e.g., misclassification and inaccurate localization), whereas the improved model (Figure 12i) achieves clearer category differentiation (color-coded boxes) and more precise positioning. This enhancement supports aerial survey robots in wide-area monitoring tasks (e.g., infrastructure inspection and disaster assessment), where accurate multi-target recognition is essential.

The specific experimental data related to the VisDrone2019 dataset in the public dataset are shown in Figure 13.

After conducting a series of experiments, it is evident that our proposed algorithm significantly improves the extraction performance for dynamic small targets compared to the original YOLOv8n algorithm. When compared to the original YOLOv8 model, the results on the public dataset indicate that our improved structure achieves approximately a 6% increase in accuracy and about a 3% improvement in both mAP0.5 and mAP0.5–0.95 metrics. Additionally, it can be observed that the improved model exhibits excellent performance in dynamic pedestrian detection and recognition, with a significant reduction in missed detections.

Table 7 presents the detailed comparison between the original YOLOv8n model and our proposed YOLOv8n-FFRD model on the VisDrone2019 dataset. All metrics are calculated following the standard evaluation protocol of the dataset.

The quantitative results in Table 7 demonstrate that the proposed YOLOv8n-FFRD model outperforms the baseline YOLOv8n in all key metrics on the VisDrone2019 dataset.

Precision is improved by 4.9%, indicating fewer false positives. This is attributed to the MLCA attention mechanism, which suppresses background interference (e.g., cluttered urban textures and vegetation) by focusing on discriminative regions of small targets (e.g., pedestrian silhouettes).

Recall increases by 2.7%, reflecting a significant reduction in missed detections. The feature fusion and rediffusion (FFRD) structure plays a critical role here; by aggregating shallow detail features (e.g., edge information of distant pedestrians) and deep semantic features (e.g., contextual relationships), it enhances the representation of small targets (≤32 × 32 pixels) that are easily overlooked in the baseline model.

mAP@0.5 andmAP@0.5–0.95 are enhanced by 2.4% and 2.9%, respectively. This validates the effectiveness of the wIoU loss function, which adaptively weights bounding box edges to improve regression accuracy—particularly valuable for dynamic targets (e.g., moving pedestrians) with unstable motion trajectories in drone-captured videos.

The consistent improvement across metrics indicates that the proposed model not only enhances small-target detection in specific environments but also possesses strong adaptability to diverse dynamic scenes, laying a foundation for its application in drone-based surveillance and mobile robot environmental perception.

## 5. Conclusions

This paper proposes a dynamic small-target detection algorithm based on a feature fusion and redistribution structure, designed specifically for mobile robots to detect dynamic small-target pedestrians and vehicles in complex dynamic environments. By introducing an attention mechanism, the model can focus more on the key regions of images, thereby improving the detection accuracy of small target objects. To verify the effectiveness of our algorithm, we conducted relevant tests in various environments and performed small-target detection experiments on multiple datasets. Experimental results show that the proposed algorithm outperforms the original YOLOv8 algorithm in detecting and recognizing small targets, especially when handling small-target pedestrians or vehicles in complex environments. Our proposed method improves accuracy by 5%, and increases mAP0.5 and mAP0.5–0.95 by approximately 3%. This improvement not only enhances the model’s ability to detect and recognize small target objects but also demonstrates its good applicability in complex environments such as nighttime conditions. Deployment experiments on portable embedded devices show that the proposed model achieves a test speed of 25 frames per second, exhibiting excellent real-time performance. This makes deployment on mobile robots easier while maintaining good performance, which is crucial for improving the robot’s environmental perception ability.

A limitation of our algorithm is that it has not undergone pruning or distillation operations to further compress the model size and make it lighter. We will investigate these areas in future work.

## Figures and Tables

**Figure 1 sensors-25-05106-f001:**
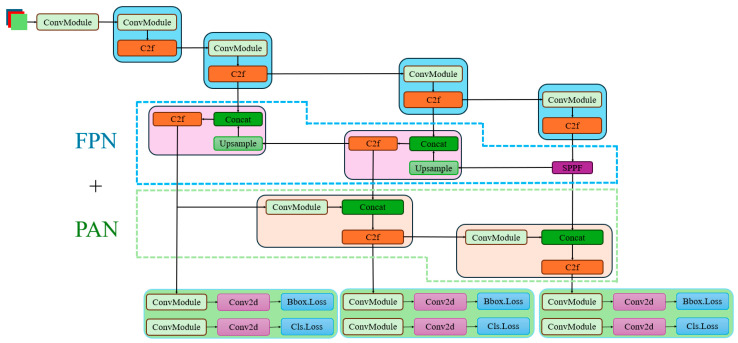
YOLOv8 basic model diagram.

**Figure 2 sensors-25-05106-f002:**
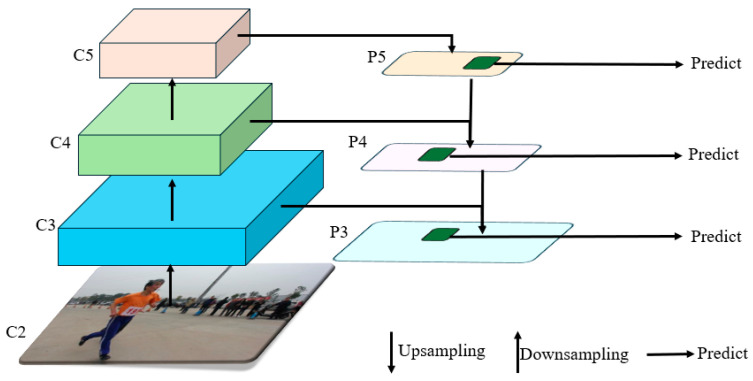
Schematic diagram of FPN-PAN structure.

**Figure 3 sensors-25-05106-f003:**
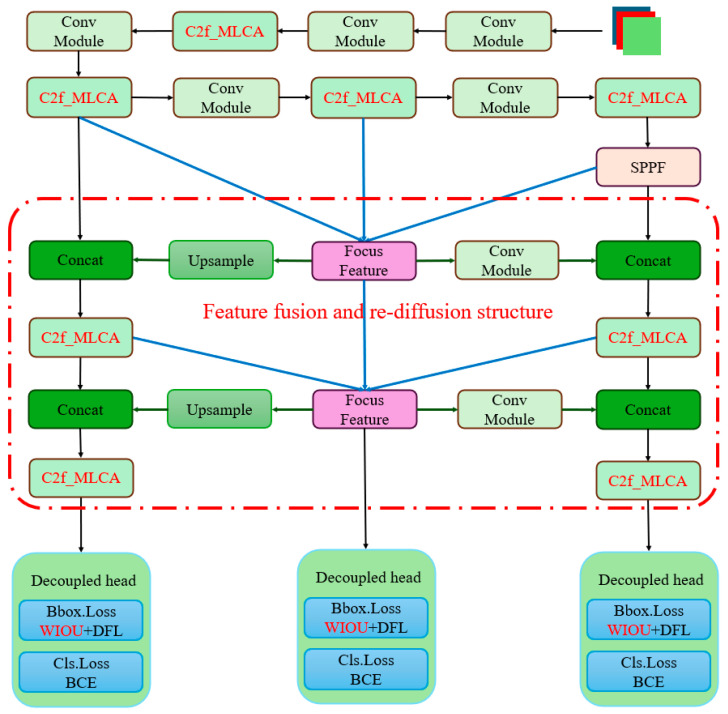
Structure diagram of improved YOLOv8 algorithm.

**Figure 4 sensors-25-05106-f004:**
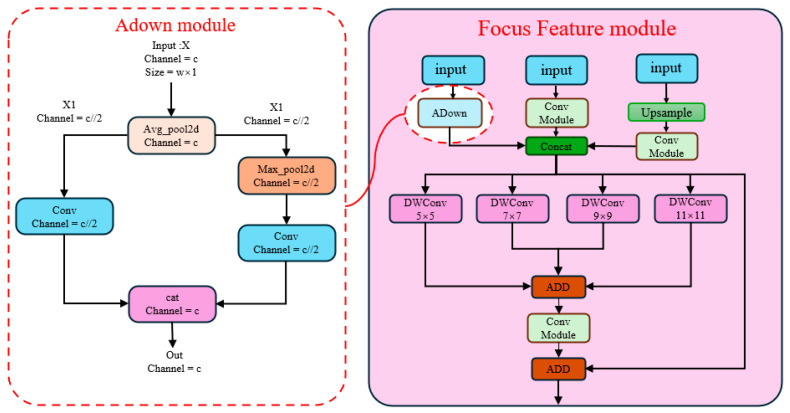
Structure diagram of the focus feature module.

**Figure 5 sensors-25-05106-f005:**
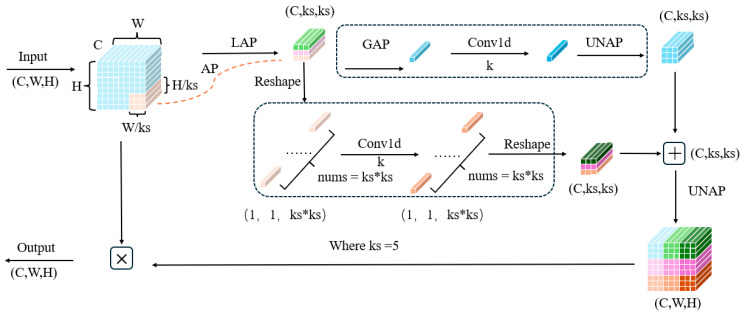
Schematic diagram of MLCA (Mixed Local Channel Attention) module.

**Figure 6 sensors-25-05106-f006:**
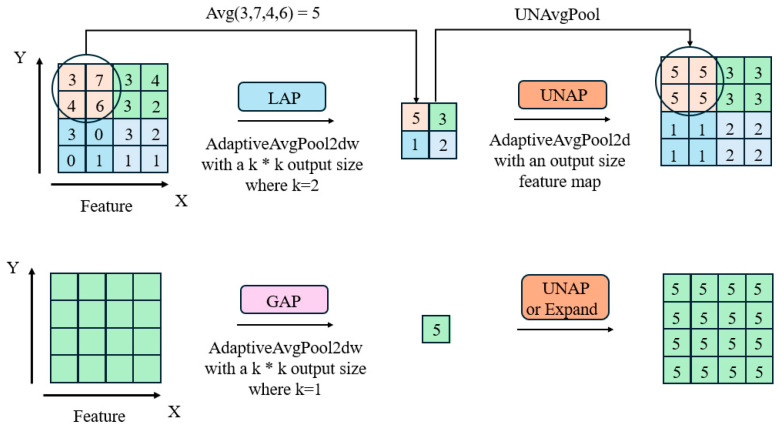
Internal operation diagram of MLCA module.

**Figure 7 sensors-25-05106-f007:**
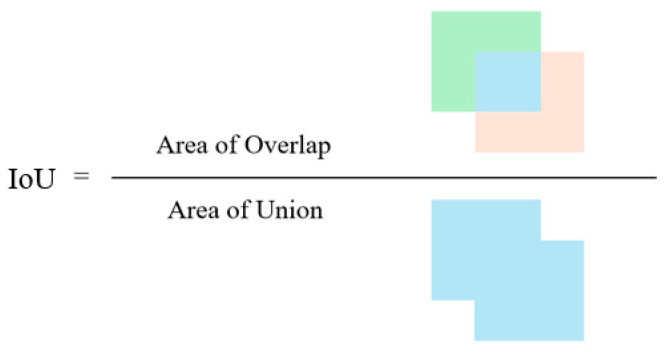
Schematic diagram of IoU calculation.

**Figure 8 sensors-25-05106-f008:**
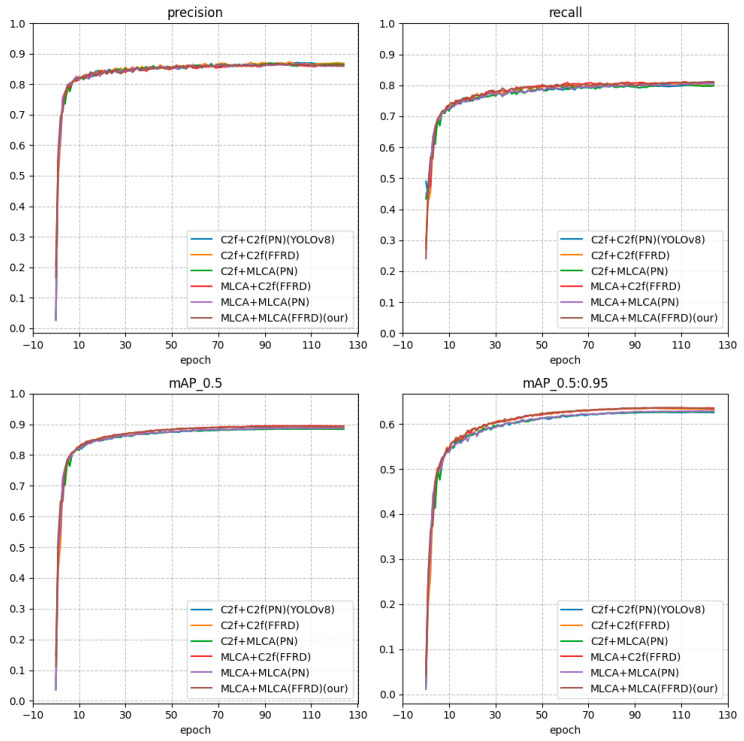
Comparison curve of key indicators.

**Figure 9 sensors-25-05106-f009:**
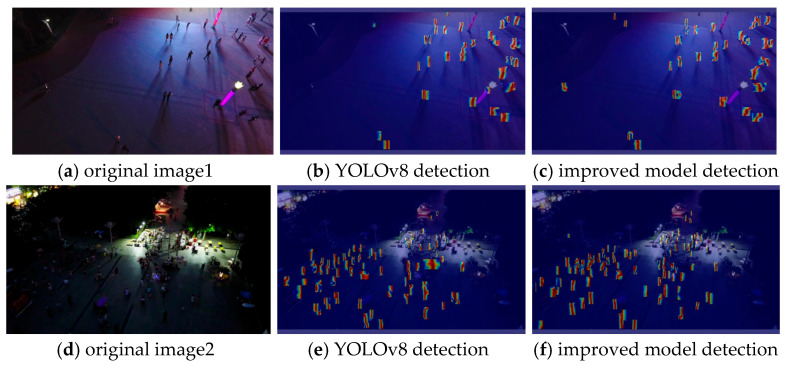
Comparison effect of the heat map, where (**a**,**d**) are the original images, (**b**,**e**) are the YOLOv8 detection heat map effects, and (**c**,**f**) are the improved model detection heat map effects.

**Figure 10 sensors-25-05106-f010:**
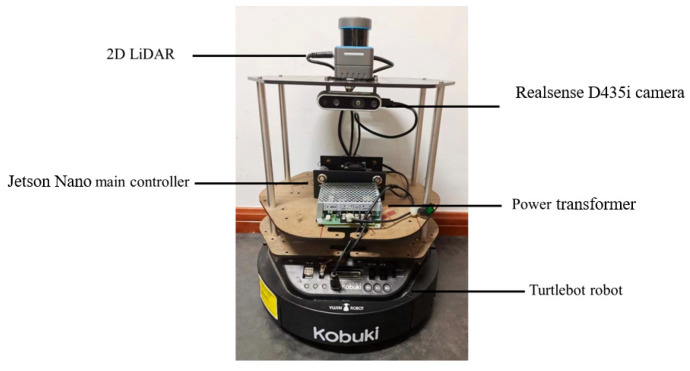
Mobile robot testing platform.

**Figure 11 sensors-25-05106-f011:**
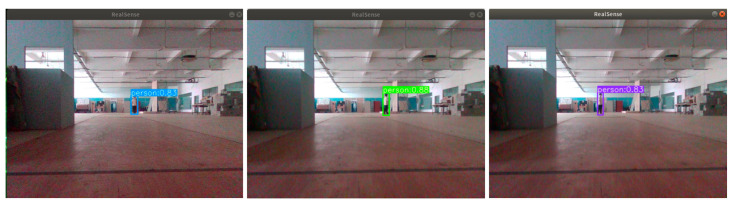
Embedded platform test results.

**Figure 12 sensors-25-05106-f012:**
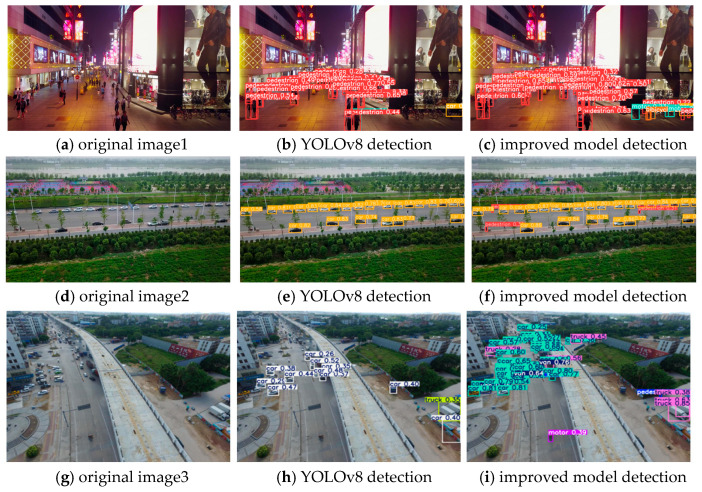
Recognition performance of VisDrone2019 public dataset, where (**a**,**d**,**g**) are the original images, (**b**,**e**,**h**) are the YOLOv8 detection effect maps, and (**c**,**f**,**i**) are the improved model detection effect maps.

**Figure 13 sensors-25-05106-f013:**
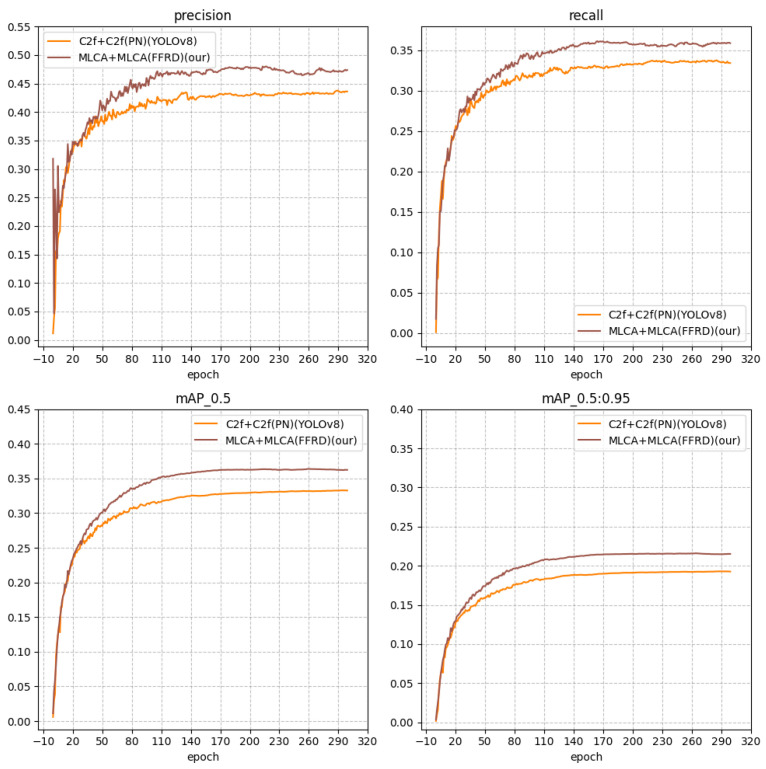
Experimental comparison of the VisDrone2019 dataset.

**Table 1 sensors-25-05106-t001:** Configuration Parameters.

Configuration Name	Version Parameters
Graphics Card (GPU)	GeForce RTX 4090 24 GB
Processor (CPU) Operating system	AMD EPYC 9654 96-Core Processor Windows 10
GPU acceleration library	Cuda 11.0
Programming language	Python 3.8.10

**Table 2 sensors-25-05106-t002:** Experimental related indicators.

Model	Precision/%	Recall/%	mAP@0.5/%	mAP@0.95/%	F1/%
C2f + C2f (PN) (YOLOv8)	86.4	79.9	88.5	62.5	82.2
C2f + C2f (FFRD)	86.5	80.5	89.0	63.0	82.5
C2f + MLCA (PN)	86.3	79.9	88.9	62.7	82.3
MLCA + C2f (FFRD)	86.3	80.6	89.5	62.6	82.8
MLCA + MLCA (PN)	84.2	80.7	89.3	62.5	82.6
MLCA + MLCA (FFRD) (OUR)	86.4	81.2	90.0	63.3	83.0

**Table 3 sensors-25-05106-t003:** Comparison of different loss functions.

*Model*	*Precision/%*	*Recall/%*	*mAP@0.5/%*	*mAP@0.5–0.95/%*
Yolov8n + CIoU	84.8	79.9	90.0	81.6
Yolov8n + DIoU	85.2	80.5	91.2	82.7
Yolov8n + GIoU	86.0	80.7	91.6	81.2
Yolov8n + wIoU (our)	86.4	81.2	92.0	83.4

**Table 4 sensors-25-05106-t004:** Algorithm comparison indicators.

*Model*	*Precision/%*	*Recall/%*	*mAP@0.5/%*	*mAP@0.95/%*	*F1/%*
YOLOv8n	86.5	79.6	89.4	62.5	82.7
SSD	85.0	77.8	87.9	62.0	82.5
MASK_RCNN	87.0	79.0	89.5	63.2	82.9
Faster-RCNN	85.2	78.9	89.2	62.9	81.4
YOLOv8n-FFRD (our)	86.7	81.5	90.1	63.5	83.2

**Table 5 sensors-25-05106-t005:** Comparison of ablation experiments.

*wIoU*	*MLCA(B)*	*MLCA(N)*	*FFRD*	*Precision/%*	*Recall/%*	*mAP@0.5/%*
×	×	×	×	85.1	79.3	87.6
×	×	×	√	85.0	79.8	88.6
×	×	√	√	85.3	80.8	88.9
√	×	√	√	85.9	80.5	89.4
√	√	×	√	86.0	80.3	89.5
√	√	√	√	86.4	81.2	90.0

**Table 6 sensors-25-05106-t006:** Performance Comparison of Different Algorithms on GPU and Embedded Platforms.

Models	mAP@0.5/%	Params/M	GFLOPs	FPS (RTX 4090)	FPS (Jetson Nano)
YOLOv8n	89.4	3.2	8.7	581	28
YOLOv8s	91	11.2	28.6	325	18
ResNet50-FPN (Faster-RCNN)	89.5	36.8	165.2	76	6
ResNet101-FPN (Faster-RCNN)	91.6	50.3	210.5	41	4
RT-DETR-S	90.2	23.4	55.8	209	-
RT-DETR-M	92.3	34.1	87.5	132	-
D0 (EfficientDet)	89.9	3.8	9.2	382	28
D7 (EfficientDet)	91.6	55.6	330.8	28	19
YOLOv8n-FFRD	90.1	2.9	7.9	631	32
YOLOv8s-FFRD	92	10.1	25.3	352	21

**Table 7 sensors-25-05106-t007:** Performance comparison on VisDrone2019 dataset.

*Model*	*Precision (%)*	*Recall (%)*	*mAP@0.5 (%)*	*mAP@0.5–0.95 (%)*	*F1 (%)*
YOLOv8n (Baseline)	43.2	33.2	33.4	19.2	37.5
YOLOv8n-FFRD (Ours)	48.1	35.9	35.8	22.1	41.1
Improvement	+4.9	+2.7	+2.4	+2.9	+3.6

## Data Availability

The original contributions presented in this study are included in the article. Further inquiries can be directed to the corresponding author..

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
