# Peer review of "YOLO-FFRD: Dynamic Small-Scale Pedestrian Detection Algorithm Based on Feature Fusion and Rediffusion Structure"

_sensors, 2025, doi:10.3390/s25165106_

Round 1
Reviewer 1 Report
Comments and Suggestions for Authors
This paper aims to propose a dynamic small target detection algorithm based on feature fusion and re-diffusion structure, suitable for deployment on mobile robotic platforms. The experimental results demonstrate that the high-performance small target pedestrian detection algorithm based on feature fusion and re-diffusion structure significantly reduces the missed detection rate and exhibits good generalization capabilities, which can be extended to multi-class small target detection.
The comments of this paper are as follows:
1. Technical issues
(1) To validate the superiority of the proposed algorithm, this paper conducts comparative experiments with various common algorithms. However, these algorithms shown in Table 4 are relatively outdated. It is recommended to perform comparative experiments with algorithms published in the past three years.
(2) In Section 4.5, the experimental results of the algorithm on the mobile robot platform are presented. However, the experimental results are insufficient. It is suggested to display the actual application effects of the algorithm on the mobile robot platform.
(3) In Section 4.6, the paper conducts experiments on the VisDrone 2019 public dataset, and the experimental results are shown in Figure 13. However, the figure fails to indicate the accurate numerical values of the improvement of the proposed model compared with the YOLOv8 model. It is advisable to present the results in a table. Moreover, the analysis and discussion in this section are not sufficient.
(4) For dynamic small-scale object detection algorithm, mode size and running speed are crucial. However, this paper did not address related issues. Therefore, the innovation is not sufficient.
2. Format issues
(1) There are instances of formatting irregularities, such as Tables 2 and 5 being split across pages.
(2) In line 166, Detection methods, etc.[27]? This sentence is not coherent.
(3) In line 281, formula 8 is missing?
(4) In line 355, "According to the experimental results presented in Table 5" should be changed to "According to the experimental results presented in Table 4".
(5) In line 395, figure 11 is missing?
(6) In line 413, "Figure 15" should be changed to "Figure 13".
(7) In lines 458-550, some references have non-standard formats.
Comments on the Quality of English LanguageThe English could be improved to more clearly express the research.
Reviewer 2 Report
Comments and Suggestions for Authors
A well written manuscript. The general level ideas , methods, results, and conclusions are well explanained so that a reader like me, not familiar with all the details of CNN implementations, can unstand the main points.
My detailed comments:
In the end of the Introduction, it is nice that list 1-4 is used to clarify the focus aspects/aims of the paper, but is the "4. experimental validation" actually a separate aspect. At least, the item 4) need to be shortened, lines 109-112 belong to discussion/conclusion section.
Please open the acronym PAN
In line 196 please open the acronym PKI (so that reader get's the overall idea of the reference [30].)
Figure 4, caption and title text in the figure are not aligned: If the large purple module on the right is the Focus Feature module, the title text "Feature fusion module" need to be replaced by "Focus Feature module". If this module is the same as the "Focus Feature" in Figure 3.
Figure 5 caption needs to explain the acronyms in the figure, or/and the caption should refer to the section 3.2 text where the concepts of the figure are explained. The asterix (*) is being used to denote multiplication. I am not used to this notation, but if it is aligned with journal style, then fine.
In equation (1) the absolute value "|" stick should to span higher and lower
In figure 7: IOU , replace with IoU
In table 1, add spaces: Cuda 11 and Python 3
line 321 is missing some words.
Acronym "C2f" is not introduced.
In figure 9 and 12, instead of only (b), c), e) and f) , please add the method acronyms to each subcaption, such as : "c) YOLOv8n-FFRD" . For readability.
Figure 9 caption should explain what the "heat"(color) indicates in the images. (For the readers who are not familiar with heat maps of this topic.)
About VisDrone2019 data set , can you describe framerate of the videoclips?
I think that one limitation of study was that the test data (VisDrone2019) was only from Chinese cities.
The point of "attention mechanism" is well explained in line 428-429. Namely, from the Method section, I didn't get that the attention mechanism is about focusing on the certain spatial regions of the image. Thus, it is preferable to clarify explanation of the attention mechanism in the method section
In describing the test data, in line 434, "complex environments" is mentioned. Does this mean complexity of the background or that there are partialocclusions of pedestrians in the photos and videos?. This needs to be explained. Especially, it would be essential to show some examples where YOLOv8n fails to detect a pedestrian, but the presented method succeeds. In the current manuscript, Figure 12 is the only where a reader can see some difference. It would be clarifying to see a few image clips where the presented method succeeds.
Round 2
Reviewer 1 Report
Comments and Suggestions for Authors
This paper has been revised according to the reviewer's comments. The formatting has been improved, additional algorithm comparison experiments have been included, and the practical application of the algorithm on a mobile robot platform has been demonstrated, with experimental results being sufficiently presented.
The additional comments of the revised paper are provided as follows:
1.In the Conclusion section, lines 529-531 state, "Deployment experiments on portable embedded devices reveal that our proposed model achieves a testing speed of 11.33 frames per second, exhibiting excellent real-time performance." However, this conclusion is not substantiated by the experimental section, where the value of 11.33 FPS is not mentioned.
2.In response to the reviewer's comments, the authors modified Point 3 and supplemented the content of Section 4.6. However, the revised manuscript does not include the relevant content. It is recommended to present specific values in the form of a table to enhance the readability of this paper.
Reviewer 2 Report
Comments and Suggestions for Authors
Thanks to the authors for detailed additions to the manuscript, especially for schematic diagrams and Figure 12 and its explanations.
These few issues need to be revised;
- Lines 89-98 and 219-222 have several extra hyphens ("ac-curacy" etc) need to be revised.
- Figure 3 is missing one hyphen (re-).
-Line 218 "which is used in accordance with the journal's formatting style." an unnecessary remark.
-Figure 12 d and g captions should have "image2" and "image3" I guess.
-Since two out of three examples in Figure 12 are on detection of small vehicles rather than pedestrians, the Conclucion and introduction section need to be aligned with these results, i.e., mentioning vehicles in addition to pedestrians in concluding remarks. (Or focusing on the pedestrians in conlcusions need to be jsutified)
For the final submisstionPlease make quality/ resolution of the figures the highest you can make by following guidance of journal and editors
